# Targeting Hypoxia and HIF1α in Triple-Negative Breast Cancer: New Insights from Gene Expression Profiling and Implications for Therapy

**DOI:** 10.3390/biology13080577

**Published:** 2024-07-31

**Authors:** Delong Han, Zeyu Li, Lingjie Luo, Hezhong Jiang

**Affiliations:** 1School of Life Science and Engineering, Southwest Jiaotong University, Chengdu 610031, China; 17883686338@163.com; 2Institute for Inheritance-Based Innovation of Chinese Medicine, Shenzhen University Medical School, Shenzhen University, Shenzhen 518055, China; l13703564582@163.com; 3Marshall Laboratory of Biomedical Engineering, Shenzhen University Medical School, Shenzhen University, Shenzhen 518055, China

**Keywords:** HIF1a, TNBC, hypoxia, mRNA-Seq, targeted therapy

## Abstract

**Simple Summary:**

Breast cancer is a leading cause of death among women worldwide, with aggressive subtypes such as triple-negative breast cancer posing a significant challenge. These cancers are often resistant to treatment and have a poor prognosis. Our research focused on understanding the role of hypoxia in the development and progression of triple-negative breast cancer. We studied the effects of hypoxia on MDA-MB-231 cells, a model for triple-negative breast cancer. Using cobalt chloride (CoCl_2_) to mimic the effects of hypoxia, we observed changes in cell behavior and gene expression. We found that hypoxia promotes the migration and survival of MDA-MB-231 cells, increasing their aggressiveness and resistance to treatment through the activation of hypoxia-inducible factor 1α. Our findings suggest that targeting hypoxia-related pathways could be a promising strategy for developing new therapies for triple-negative breast cancer. By understanding how hypoxia contributes to the aggressive nature of these cancers, we can develop more effective treatments that will improve patient outcomes. This research contributes to the development of new and more effective treatments for aggressive breast cancer. By targeting hypoxia, we aim to improve the survival rates and quality of life of patients with this devastating disease.

**Abstract:**

Breast cancer is a complex and multifaceted disease with diverse risk factors, types, and treatment options. Triple-negative breast cancer (TNBC), which lacks the expression of estrogen receptor, progesterone receptor, and human epidermal growth factor receptor 2 (HER2), is the most aggressive subtype. Hypoxia is a common feature of tumors and is associated with poor prognosis. Hypoxia can promote tumor growth, invasion, and metastasis by stimulating the production of growth factors, inducing angiogenesis, and suppressing antitumor immune responses. In this study, we used mRNA-seq technology to systematically investigate the gene expression profile of MDA-MB-231 cells under hypoxia. We found that the hypoxia-inducible factor (HIF) signaling pathway is the primary pathway involved in the cellular response to hypoxia. The genes in which expression levels were upregulated in response to hypoxia were regulated mainly by HIF1α. In addition, hypoxia upregulated various genes, including *Nim1k*, *Rimkla*, *Cpne6*, *Tpbgl*, *Kiaa11755*, *Pla2g4d*, and *Ism2*, suggesting that it regulates cellular processes beyond angiogenesis, metabolism, and known processes. We also found that HIF1α was hyperactivated in MDA-MB-231 cells under normoxia. A HIF1α inhibitor effectively inhibited the invasion, migration, proliferation, and metabolism of MDA-MB-231 cells. Our findings suggest that hypoxia and the HIF signaling pathway play more complex and multifaceted roles in TNBC than previously thought. These findings have important implications for the development of new therapeutic strategies for TNBC.

## 1. Introduction

Breast cancer is the most common cancer among women worldwide and is a leading cause of cancer-related death. In 2018 and 2023, the Global Cancer Observatory reported over 2 million and approximately 3 million cases, respectively [1,2]. Breast cancer is a complex and multifaceted disease with diverse risk factors, types, and treatment options [3]. Breast cancer can be categorized into three major subtypes: hormone receptor-positive/*ERBB2*-negative, *ERBB2*-positive, and triple-negative [4]; triple-negative breast cancer (TNBC) is the most aggressive subtype.

TNBC encompasses a heterogeneous group of fundamentally different diseases with distinct histologic, genomic, and immunologic profiles, which are aggregated under this term because of their lack of estrogen receptor, progesterone receptor, and human epidermal growth factor receptor 2 expression [5]. Current treatment options for breast cancer include surgery, radiation therapy, chemotherapy, hormonal therapy, targeted therapy, and immunotherapy [6,7]. The optimal treatment strategy for breast cancer depends on the tumor subtype, stage, and individual patient characteristics [8]. Prolonging life and palliating symptoms are the main goals for patients with metastatic breast cancer [8]. For nonmetastatic breast cancer, endocrine therapy is used mainly for patients with hormone receptor-positive tumors, and a minority of patients also receive chemotherapy [9]; *ERBB2*-targeted antibody or small-molecule inhibitor therapy, combined with chemotherapy, is used mainly for patients with *ERBB2*-positive tumors [10]; patients with triple-negative tumors receive chemotherapy alone [10]. Local therapy for all patients with nonmetastatic breast cancer consists of surgical resection, with the consideration of postoperative radiation if a lumpectomy is performed [11]. Additionally, targeted therapies, particularly those targeting epidermal growth factor receptor (EGFR), poly (ADP-ribose) polymerase 1 (PARP1), vascular endothelial growth factor (VEGF), cyclin-dependent kinases 4 and 6 (CDK4/6), and phosphatidylinositol-3-kinase (PI3K), have significantly improved the survival rates of patients with breast cancer [12]. However, targeted therapies are prone to developing resistance, leading to treatment failure. Resistance can be categorized into intrinsic and acquired resistance. Intrinsic resistance includes genetic mutations, the activation of defense pathways, and cancer stem cell activation. Acquired resistance mechanisms include oncogene activation, changes to the tumor microenvironment [13], epigenetic modifications, enhanced DNA damage repair, and gene expression changes due to mutations [14]. For a detailed review of drug resistance mechanisms, please refer to the comprehensive article by Rajesh et al. [14]. While these treatments increase the two- and five-year survival rates for women with breast cancer, they cannot cure the disease. Therefore, it is important to explore new mechanisms and drugs for treating breast cancer.

The tumor microenvironment (TME) is a complex ecosystem that surrounds and interacts with a tumor [15]. It is composed of a variety of cell types, including cancer cells, immune cells, stromal cells, and endothelial cells, as well as the extracellular matrix (ECM) and soluble factors [16]. The TME plays a critical role in tumor development and progression. It can promote tumor growth [17], invasion [18], and metastasis [19,20], and it can also suppress antitumor immune responses [21]. Microenvironmental hypoxia, a common feature of solid tumors, plays an important role in tumor development and progression [22]. It occurs when tumor growth is rapid, leading to abnormal blood vessels and increased oxygen consumption.

Hypoxia can promote breast tumor growth by stimulating the production of growth factors and inhibiting the production of tumor suppressors [23]. It can also induce angiogenesis and the formation of new blood vessels, which improves oxygen delivery to tumors and promotes their growth [24]. Furthermore, hypoxia can promote tumor invasion and metastasis by stimulating the production of matrix metalloproteinases (MMPs), which are enzymes that break down the ECM [25,26]. Hypoxia also regulates migration/invasion and cell death by modulating the ion channels, such as the Ca^2+^, Cl^−^, and K^+^ channels [27,28,29]. Finally, hypoxia can suppress antitumor immune responses by inhibiting the function of immune cells [30]. Thus, targeting hypoxia is a promising approach for cancer therapy.

Although hypoxia promotes the development of TNBC and is involved in multiple processes, no current drugs target these processes to cure TNBC. Thus, it is important to explore new mechanisms of TNBC under hypoxia and screen for new drugs that target the hypoxia signaling pathway. Here, we demonstrate that a hypoxia-inducing factor (HIF)1α inhibitor can effectively inhibit the expression of genes associated with angiogenesis, invasion, glycolysis, and EMT. We also identified several new markers of the TNBC response to hypoxia, including *Nim1k*, *Rimkla*, *Cpne6*, *Tpbgl*, *Kiaa11755*, *Pla2g4d*, and *Ism2*. These new markers are also regulated by HIF1α.

## 2. Materials and Methods

### 2.1. Cell Culture

MDA-MB-231 cells were maintained in Dulbecco’s modified Eagle’s medium (DMEM), supplemented with 10% fetal bovine serum (FBS), and incubated at 37 °C and 5% CO_2_ under 1% O_2_ for hypoxia or 21% O_2_ for normoxia experiments. For HIF-1α-IN-2 (MCE, Monmouth Junction, NJ, USA, cat. no. HY-115903)-mediated inhibition, MDA-MB-231 cells were incubated at 37 °C, 5% CO_2_, and 21% O_2_ and treated with the indicated concentrations.

### 2.2. ROS Detection

MDA-MB-231 cells were incubated with 10 µM 2′,7′-dichlorofluorescin diacetate (DCFDA) in FBS-free DMEM for 20 min after overnight growth and then washed 3 times with PBS. Levels of 2′,7′-dichlorofluorescein (DCF) were measured with a Cytation1 instrument (Bio-Tek; Winooski, VT, USA) with spectra of 469 nm excitation/525 nm emission. The fluorescence intensity was measured using ImageJ 8.

### 2.3. Cell Viability Assay

Cell viability was measured with a CCK-8 assay kit (Meilunbio^®^, Dalian, China, cat. no. MA0218) according to the manufacturer’s instructions. MDA-MB-231 cells were treated with CoCl_2_ (Macklin, Shanghai, China, cat. no. C885204) for the indicated times after overnight growth. One hundred microliters of working solution (FBS-free DMEM:CCK-8 solution = 9:1) was added to each well of a 96-well plate and incubated for an additional 0.5–2 h. The absorbance at 450 nm was analyzed with a Cytation 1 instrument (Bio-Tek; Winooski, VT, USA).

### 2.4. In Vitro Migration Assay

Images of MDA-MB-231 cell migration into wound regions were monitored at 0 h and 48 h (T0 and T48) with a Cytation 1 instrument (Bio-Tek; Winooski, VT, USA). The wound closure rate was calculated as follows: % scratch = (width at T0 − width at T48)/width at T0 × 100. The data were analyzed with the Cytation 1 and Prism 8 software. For the invasion migration assay, MDA-MB-231 cell migration was evaluated in Transwell chambers (Corning, New York, NY, USA, cat. no. 3422) according to the manufacturer’s protocol. MDA-MB-231 cells were fixed and stained with 1% crystal violet (Solarbio, Beijing, China, cat. no. C8470), then washed, dried, and imaged after transfection for 12 h. The number of cells was determined using ImageJ 8.

### 2.5. Western Blot Analysis

MDA-MB-231 cells were lysed with a radioimmunoprecipitation assay (RIPA) lysis buffer (Beyotime, Nantong, China, cat. no. P0013B), which included proteinase inhibitors (MIKX, Shenzhen, China, cat. no. DB612A) and phosphatase inhibitors (MIKX, cat. no. DB615). Equal amounts of protein were separated on a 10% SDS-PAGE gel and transferred to an NC membrane (Millipore, Billerica, MA, USA) after centrifugation at 12,000 rpm for 15 min at 4 °C. The samples were blocked with 5% BSA in PBST for 1 h at room temperature and then incubated with primary antibodies (Vimentin (1:1000; rabbit; cat. no. 5741T, CST, Danvers, MA, USA), Snail (1:1000; rabbit; cat. no. 3879T, CST). β-actin (1:5000; mouse; cat. no. YM-3028, Immunoway, Plano, TX, USA). HIF-1α (1:1000; rabbit; cat. no. NB 100-479, NOVUS, Saint Charles, MO, USA)) and secondary antibodies. The membranes were developed for chemiluminescence detection using an enhanced chemiluminescence (ECL) detection kit (MIKX, cat. no. MK-S700) [31]. The band intensities were quantified using ImageJ 8.

### 2.6. Quantitative Real-Time PCR Analysis

First, 1 μg of RNA was reverse transcribed to cDNA by reverse transcriptase (Vazyme, Nanjing, China, cat. no. R323) after removing the genomic DNA. Then, qPCR was performed in triplicate with SYBR-Green PCR Master Mix (Vazyme, cat. no. Q712). The primers used in this study were as follows: HsGlut1 forward, 5′-GGCCAAGAGTGTGCTAAAGAA-3′ and reverse, 5′-ACAGCGTTGATGCCAGACAG-3′; HsLdha forward, 5′-ATGGCAACTCTAAAGGATCAGC-3′ and reverse, 5′-CCAACCCCAACAACTGTAATCT-3′; HsPdk1 forward, 5′-CTGTGATACGGATCAGAAACCG-3′ and reverse, 5′-TCCACCAAACAATAAAGAGTGCT-3′; HsVegf forward, 5′-AGGGCAGAATCATCACGAAGT-3′ and reverse, 5′-AGGGTCTCGATTGGATGGCA-3′; HsNim1k forward, 5′-GATCGGCGCGACAGTGTAG-3′ and reverse, 5′-CAGCGTGATCTCCCTCACC-3′; HsRimkl forward, 5′-GGATGTGGTGCTTGTACGGG-3′ and reverse, 5′-AAGATGCTCTGTGGGCGATTG-3′; HsIsm2 forward, 5′-CTAACCCTGATACCCAGGCTT-3′ and reverse, 5′-CAGCAAGGGAGTAACCTCCT-3′; HsTpbgl forward, 5′-GCCTCATCTTCCTCATGGT-3′ and reverse, 5′-CTCGTAGCGGTAGTGGTAG-3′; HsCpne6 forward, 5′-TGAGCGAGTTCGACTCCTTG-3′ and reverse, 5′-GTGGGTCGGACTTGGAAAACA-3′; HsPla2g4d forward, 5′-GCTGACCTGTTGAGTGAGGC-3′ and reverse, 5′-TCGGTGAGCGTCTTGGTCTTA-3′; Kiaa1755 forward, 5′-CTGGGTCAGGTGTTCCGTC-3′ and reverse, 5′-GCAGGGATCAGGAAATCCAGA-3′. All values were normalized to the expression levels of β-actin. Relative quantification was determined using the comparative Ct method [31].

### 2.7. RNA-Seq and Data Analysis

MDA-MB-231 cells that had been under 1% O_2_ for hypoxia or 21% O_2_ for normoxia for 24 h were collected. The methods used for total RNA extraction and RNA-seq were the same as previously described [32]; briefly, RNA integrity was assessed using an RNA Nano 6000 Assay Kit for the Bioanalyzer 2100 system (Agilent Technologies, Santa Clara, CA, USA) [33]. Then, the mRNA was purified, and first-strand cDNA was synthesized. Second-strand cDNA synthesis was subsequently performed using DNA Polymerase I and RNase H [34]. After adenylation of the 3′ ends of the DNA fragments, adaptors with hairpin loop structures were ligated to prepare for hybridization. The library fragments were purified with the AMPure XP system [35]. Then, PCR was performed with Phusion High-Fidelity DNA polymerase, universal PCR primers, and index (X) primers. Finally, the PCR products were purified (AMPure XP system, Brea, CA, USA), and library quality was assessed with an Agilent Bioanalyzer 2100 system. Clustering of the index-coded samples was performed on a c Bot Cluster Generation System using the TruSeq PE Cluster Kit v3-cBot-HS (Illumina, San Diego, CA, USA) according to the manufacturer’s instructions [36]. After cluster generation, the library preparations were sequenced on an Illumina Nova seq platform, and 150 bp paired-end reads were generated.

The raw data (raw reads) in fastq format were first processed through the fastp software. In this step, clean data (clean reads) were obtained by removing reads containing adapters, reads containing poly-N sequences, and low-quality reads from the raw data. The index of the reference genome was built using HISAT2 v2.0.5 and paired-end clean reads were aligned to the reference genome using HISAT2 v2.0.5 [37]. The mapped reads of each sample were assembled using String Tie (v1.3.3b) in a reference-based approach [38]. Feature Counts v1.5.0-p3 was used to count the number of reads mapped to each gene [39]. The FPKM value of each gene was subsequently calculated, based on the length of the gene and the number of reads mapped to that gene. Differential expression analysis and gene ontology (GO) and Kyoto Encyclopedia of Genes and Genomes (KEGG) enrichment analyses of the DEGs were implemented in the cluster Profiler R package, in which gene length bias was corrected [40,41,42]. GO terms with corrected *p*-values of less than 0.05 were considered significantly enriched in the DEGs. We have also deposited the raw data from our RNA-sequencing experiments in the GEO database (PRJNA1132955).

## 3. Results

### 3.1. Establishment of a Hypoxia Model in MDA-MB-231 Cells

To explore the potential links between hypoxia and breast cancer, we established hypoxia- and CoCl_2_-induced MDA-MB-231 cell models. Compared with those under normoxia, the expression levels of *Pdk1*, *Glut1*, *Vegf*, and *Ldha* were significantly increased in hypoxia-induced MDA-MB-231 cells at 24 h (Figure 1A). The expression levels of *Glut1*, *Vegf*, and *Ldha* were increased in CoCl_2_-treated MDA-MB-231 cells at different time points (Figure 1B). Furthermore, we found that CoCl_2_ treatment could activate HIF1α (Figure 1C,D), inhibit proliferation (Figure 1E), and induce ROS production and apoptosis (Figure 1F,G; Appendix A), which is similar to hypoxia induction in MDA-MB-231 cells. However, CoCl_2_ treatment inhibited invasion and migration, which contradicts the high invasiveness and migration of MDA-MB-231 cells under hypoxia (Figure 1H–M). These results indicate that CoCl_2_ can only partially mimic hypoxia induction. The expression profiles of genes under normoxia (*n* = 2) and hypoxia (*n* = 2) were separately characterized through mRNA analysis.

### 3.2. Transcriptomic Profiles of Normoxic and Hypoxic MDA-MB-231 Cells Reveal Common and Divergent Transcriptomic Patterns

To investigate the effects of hypoxia on gene expression and regulation in MDA-MB-231 cells, we used mRNA-Seq to analyze the transcriptomes of MDA-MB-231 cells under normoxia (*n* = 2) and hypoxia (*n* = 2). A volcano plot created from the data revealed that 411 genes were significantly upregulated in the hypoxia group, compared with the normoxia group, and 83 genes were significantly downregulated (Figure 2A). A heatmap was generated and demonstrated significant differences in gene expression profiles between the hypoxia and normoxia groups (Figure 2B). Gene ontology (GO) analysis showed that the upregulated differentially expressed genes (DEGs) were associated with various biological processes, including oxidoreductase activity, signaling receptor binding, receptor regulator activity, receptor ligand activity, isomerase activity, and oxidoreductase activity, acting on paired donors with the incorporation or reduction of molecular oxygen (Figure 2C). Kyoto Encyclopedia of Genes and Genomes (KEGG) pathway analysis showed that the upregulated DEGs were involved in pathways such as the HIF-1 signaling pathway, glycolysis/gluconeogenesis, fructose and mannose metabolism, the biosynthesis of amino acids, carbon metabolism, microRNAs in cancer, the PI3K-Akt pathway, the MAPK pathway, and the VEGF pathway (Figure 2D).

GO analysis of the downregulated differentially expressed genes (DEGs) showed that they were associated with the immune response, immune system process, DNA replication, DNA metabolic process, extracellular region, cytokine receptor binding, chemokine activity, chemokine receptor binding, G protein-coupled receptor binding, cytokine activity, signaling receptor binding, receptor regulator activity, and receptor ligand activity (Figure 2E). The results of the KEGG analysis showed that the downregulated DEGs were involved in the TNF signaling pathway, the IL-17 signaling pathway, the cytokine-cytokine receptor interaction, the NF-kappa B signaling pathway, the cell cycle, glutathione metabolism, the NOD-like receptor signaling pathway, the JAK-STAT signaling pathway, the chemokine signaling pathway, and the p53 signaling pathway (Figure 2F). These findings suggest that hypoxia plays a role in regulating metabolic reprogramming and immune responses.

### 3.3. HIF1α Regulated the Responses of Most Genes to Hypoxia

To further explore the transcription factors (TFs) that regulate genes responsive to hypoxia, we analyzed the TFs of the 411 genes in which the expression levels were upregulated under hypoxia and found that these genes were regulated mainly by HIF1a and SP1 (Table 1). Using the HIF1α inhibitor HIF-1α-IN-2, we effectively reduced the mRNA expression levels of *Pdk1*, *Glut1*, *Vegf*, and *Ldha* induced by hypoxia (Figure 3A), reduced ROS production (Figure 3B,C), and inhibited the migration of MDA-MB-231 cells under hypoxia (Figure 3D,E). HIF-1α-IN-2 also reduced the mRNA expression levels of *Pdk1, Glut1, Vegf*, and *Ldha* (Figure 3F), decreased ROS production (Figure 3G,H), and reduced the protein levels of Vimentin and Snail, which are involved in the EMT (Figure 3I,J). Interestingly, we detected a high protein level of HIF1α in MDA-MB-231 cells under normoxia (Figure 3I,J). Thus, we treated MDA-MB-231 cells with HIF-1α-IN-2 under normoxia and found that HIF-1α-IN-2 efficiently inhibited the proliferation (Figure 3K), invasion (Figure 3L,M), and migration of MDA-MB-231 cells (Figure 3N,O). These results indicate that HIF1a is an efficient target for TNBC treatment.

We also analyzed those genes in which the expression levels were downregulated under hypoxia. The TFs of these downregulated genes were regulated mainly by SP1, E2F1, RELA, and NFKB1 (Table 2). Notably, most of these TFs, such as EGR2, JUND, E2F1, HMGA1, RELA, and NFKB1, are associated with immune responses. These findings indicate that immune responses are inhibited under hypoxia, further suggesting that hypoxia creates an immunosuppressive environment that allows tumors to evade immune surveillance and grow unchecked.

### 3.4. New Genes Related to the Hypoxia Response of MDA-MB-231 Cells

To further explore the genes responsive to hypoxia, we analyzed those genes in which expression levels were upregulated under hypoxia. In addition to the known hypoxia-inducible genes, we identified several novel hypoxia-inducible genes, *Nim1k*, *Rimkla*, *Cpne6*, *Tpbgl*, *Kiaa1755*, *Pla2g4d*, and *Ism2*. The qPCR results further showed that the mRNA expression levels of these genes increased under hypoxia (Figure 4A). The expression levels of *Rimkla* and *Nim1k* increased under CoCl_2_ treatment, whereas the expression levels of *Cpne6*, *Tpbgl*, and *Ism2* decreased (Figure 4B). Nim1k is a serine/threonine protein kinase; serine/threonine protein kinases play important roles in signal transmission under hypoxia, such as the PI3K/AKT signal transduction pathway [43] and the mTOR signal transduction pathway [44]. Thus, Nim1k may participate in signal transmission under hypoxia. Rimkla is a ribosomal modification protein rimK-like family member A that is a NAAG synthetase (NAAGS-II); NAAG protects against injury induced by NMDA and hypoxia [45,46]. Cpne6 is a calcium-binding protein that can trigger calcium signals by activating and recruiting the Rho GTPase Rac1 to cell membranes [47]. In breast cancer cells, the regulation of [Ca^2+^] is crucial for tumorigenesis and the development of cancer hallmarks, including cell growth and proliferation, migration, metastasis, and resistance to apoptosis [48,49]. Cpne6 may play an important role in regulating Ca^2+^ influx. *Tpbgl* is predicted to be involved in the negative regulation of the canonical Wnt signaling pathway. Research has indicated that the high expression of Kiaa1755 is present in breast cancer patients [50], but our results revealed that the expression of Kiaa1755 is upregulated in MDA-MB-231 cells under hypoxia, which finding should be further studied in the future. Pla2g4d belongs to the phospholipase A2 (PLA2) enzyme family, which catalyzes the hydrolysis of glycerophospholipids at the sn-2 position and then liberates arachidonic acid (AA) and lysophospholipids [51]. PLA2 is involved in multiple diseases, such as cardiovascular disease [52], Parkinson’s disease [53], and Alzheimer’s disease [54]. The high expression of PLA2 indicates the malignant potential of human breast cancer [55,56]. The function of Ism2 is not clear, and the role of Ism2 in MDA-MB-231 cells remains to be further studied.

Furthermore, to verify the TFs of these genes, we detected their mRNA expression levels under hypoxia in the presence of the HIF-1α inhibitor HIF-1α-IN-2. The qPCR results showed that the expression levels of *Nim1k*, *Rimkla*, *Tpbgl*, *Kiaa11755*, *Pla2g4d*, and *Ism2* decreased after HIF-1α-IN-2 treatment, indicating that these genes are novel HIF1α-regulated genes (Figure 4C,D). We also found that the expression levels of *Rimkla* and *Nim1k* decreased after HIF-1α-IN-2 treatment in CoCl_2_-treated MDA-MB-231 cells (Figure 4D).

## 4. Discussion

TNBC is a highly aggressive subtype of breast cancer [57]. Hypoxia is a common feature of solid cancer and is involved in multiple biological processes, including proliferation, invasion, anti-apoptosis, migration, immunosuppression, and metabolism. Therefore, exploring novel mechanisms of hypoxia and identifying new drugs that target hypoxia-related pathways are promising therapeutic approaches for TNBC. In this study, we systematically explored the gene expression profiles of MDA-MB-231 cells under hypoxia. We found that HIF1α is highly expressed in MDA-MB-231 cells and that its expression increases under hypoxia. Furthermore, the HIF signaling pathway was identified as the main pathway involved in the response to hypoxia.

CoCl_2_ is a chemical compound that has been widely used as a hypoxia mimetic in research studies [58]. CoCl_2_ mimics the effects of hypoxia by stabilizing the hypoxia-inducible factor, which plays a critical role in the cellular response to low oxygen levels. Hypoxia has been implicated in the development and progression of TNBC. Hypoxic conditions promote tumor growth, angiogenesis, and metastasis. Additionally, hypoxia contributes to resistance to chemotherapy and radiotherapy. CoCl_2_ can be a valuable tool for studying the role of hypoxia in TNBC. By treating TNBC cells with CoCl_2_, researchers can investigate the effects of hypoxia on cell proliferation, survival, migration, and invasion [59]. These studies can provide insights into the mechanisms by which hypoxia contributes to TNBC progression and help identify potential therapeutic targets. However, CoCl_2_ has been shown to have effects that are independent of HIF, which can complicate the interpretation of results obtained using this compound. CoCl_2_ has been reported to inhibit the proliferation of TNBC cells, and a combined treatment of glibenclamide and CoCl_2_ inhibits growth in highly metastatic breast cancer [60].

Therefore, we compared the similarities and differences between hypoxic and CoCl_2_-treated MDA-MB-231 cells. Our results indicate that hypoxia and CoCl_2_ treatment have distinct effects on MDA-MB-231 cells. Hypoxia has been reported to promote the invasion and migration of breast cancer cells, whereas CoCl_2_ treatment inhibited invasion and migration in our model system. These findings suggest that CoCl_2_ treatment can only partially mimic hypoxia in MDA-MB-231 cells (Figure 1). The different gene expression patterns of *HsTbpgl*, *HsCpne6*, and *HsIsm2* between hypoxia and CoCl_2_ treatment further support this conclusion (Figure 3A,B). Our study did not include RNA-sequencing analysis on CoCl_2_-treated cells for comparison with cells exposed to actual hypoxia. This comparison would have provided valuable insights into the similarities and differences between CoCl_2_-induced and hypoxia-induced gene expression changes in MDA-MB-231 cells. Future studies should aim to address this limitation by conducting RNA-sequencing analysis on both CoCl_2_-treated and hypoxic MDA-MB-231 cells. This study provides a more comprehensive understanding of the effects of CoCl_2_ on gene expression and its potential as a tool for studying hypoxia-related mechanisms in triple-negative breast cancer research.

TNBC lacks the hormone receptors estrogen receptor (ER), progesterone receptor (PR), and human epidermal growth factor receptor 2 (HER2), which are common targets of endocrine therapies and targeted therapies [61]. In addition, TNBC is highly heterogeneous and comprises a group of cells with different molecular characteristics [62]. This heterogeneity makes it difficult to develop a single therapy that targets all TNBC cells. Radiotherapy and chemotherapy are traditional treatment methods for TNBC. However, the hypoxic microenvironment can lead to chemo- and radio resistance. Immunotherapy also faces challenges due to hypoxia-induced immunosuppression [63,64]. Our results further demonstrated that immune-associated genes, which are regulated by SP1, E2F1, RELA, and NFKB1, are downregulated under hypoxia (Table 2). These findings indicate that immunosuppression occurs in MDA-MB-231 cells under hypoxia.

Recent research has indicated that hypoxia can induce multiple processes involved in TNBC development, such as invasion, anti-apoptosis, migration, and metabolism [22,65,66]. HIF1α is a key TF that regulates gene expression in the hypoxic signaling pathway. This pathway is a cellular response to decreased oxygen levels. When oxygen levels decrease, HIF1α is stabilized and dimerizes with the HIF1β subunit to form the active HIF1 transcription factor complex [67,68,69]. The HIF1 transcription factor complex then binds to the hypoxia response elements (HREs) in DNA, activating the transcription of numerous genes [70]. These genes are involved in various processes, including glycolysis [71], angiogenesis [72], invasion [73], metastasis [74], and immune escape [75], promoting TNBC survival and development. Thus, targeting HIF1α is a promising method for treating TNBC. Our results showed that a HIF1α inhibitor could efficiently inhibit ROS production and reduce the mRNA expression of the angiogenesis-associated gene *HsVegf* and the glycolysis-associated genes *HsLdha*, *HsPdk1*, and *HsGlut1*, and decrease the protein levels of the EMT-associated proteins Snail and Vimentin (Figure 3). These results further indicate that HIF1α is an important target for TNBC treatment. HIF2α is a crucial player in the hypoxic response and could have significant implications for TNBCs [22,76,77]. Thus, we compared the FPKM values of *HIF1α* and *HIF2α* (Appendix A) and found that *HIF-1α* mRNA was upregulated fourfold under hypoxic conditions, whereas *HIF2α* mRNA did not appear to be upregulated in our RNA-sequencing data. Kyoto Encyclopedia of Genes and Genomes (KEGG) pathway analysis revealed that the upregulated DEGs were involved in pathways related mainly to the HIF-1 signaling pathway (Figure 2D). The TFs of the 411 genes in which expression levels were upregulated under hypoxia were regulated mainly by HIF1a and SP1 (Table 1). The regulatory mechanism of the MDA-MB-231 cell line used in our study may differ from those of other TNBC cell lines. We will investigate the potential role of HIF2α in mediating the effects of hypoxia on MDA-MB-231 cells in the future.

Through mRNA-seq technology, we identified several novel genes that are upregulated in MDA-MB-231 cells under hypoxia, including *Nim1k*, *Rimkla*, *Cpne6*, *Tpbgl*, *Kiaa11755*, *Pla2g4d*, and *Ism2*. These results suggest the involvement of previously unknown processes in hypoxia, potentially offering new therapeutic targets for TNBC and further clarifying its heterogeneity. Our results indicate that hypoxia may regulate a wider range of cellular processes in TNBC cells than previously thought. This could lead to the development of novel therapeutic strategies targeting these additional pathways.

HIF1α is overexpressed in more than 80% of TNBC cases and is hyperactivated in TNBC cells [66]. In our study, we found that HIF1α was activated in MDA-MB-231 cells under normoxia and that a HIF1α inhibitor effectively inhibited the proliferation, migration, and invasion of MDA-MB-231 cells (Figure 3). These results further clarified that HIF1α is a potential therapeutic target for TNBC. In addition, the high expression of HIF1α in MDA-MB-231 cells under normoxia suggests that MDA-MB-231 cells have evolved mechanisms to bypass typical oxygen-sensing pathways. These findings indicate that the HIF1α signaling pathway may be activated by other mediators. Indeed, multiple signaling pathways, such as the NF-κB, RAS-RAF-MEK-ERK, PI3K/Akt/mTOR, and JAK-STAT pathways, regulate the expression of HIF1α [66]. In addition to HIF1α, ion channels (K^+^ and Cl^−^) can lead to abnormal migration/invasion and resistance to cell death under hypoxia [27,28,29]. Therefore, exploring the abnormal mechanisms of oxygen-sensing pathways and the hyperactivated signaling pathways leading to HIF1α accumulation is crucial.

Our results revealed that an HIF1α inhibitor could efficiently inhibit ROS production; inhibit the proliferation and migration of MDA-MB-231 cells; reduce the mRNA expression of the angiogenesis-associated gene *HsVegf* and the glycolysis-associated genes *HsLdha, HsPdk1*, and *HsGlut1*; and decrease the protein levels of the EMT-associated proteins Snail and Vimentin. These results suggest that HIF-1α-IN-2, a potent and selective inhibitor of HIF1α, has promising potential for clinical applications in the treatment of TNBC. HIF-1α-IN-2 could be used in combination with radiotherapy and chemotherapy to enhance their efficacy and overcome resistance. In addition, HIF-1α is overexpressed in many other types of cancer, including clear cell renal cell carcinoma, head and neck cancer, and pancreatic cancer. Moreover, HIF-1α-IN-2 could be used to treat these cancers. Thus, HIF-1α-IN-2 is a promising new therapeutic agent with potential applications in the treatment of TNBC and other cancers. Further research is needed to fully evaluate its clinical potential and to develop optimal treatment strategies.

## 5. Conclusions

In summary, our results indicate that the HIF signaling pathway is the primary pathway involved in the response to hypoxia in MDA-MB-231 cells. Targeting HIF1α effectively inhibits the invasion, migration, proliferation, and metabolism of MDA-MB-231 cells. In addition, we identified novel marker genes that are upregulated in MDA-MB-231 cells under hypoxia and that are regulated by HIF1α, including *Nim1k*, *Rimkla*, *Cpne6*, *Tpbgl*, *Kiaa11755*, *Pla2g4d*, and *Ism2.* Our data also suggest that CoCl_2_ can only partially mimic hypoxia in MDA-MB-231 cells.

## Figures and Tables

**Figure 1 biology-13-00577-f001:**
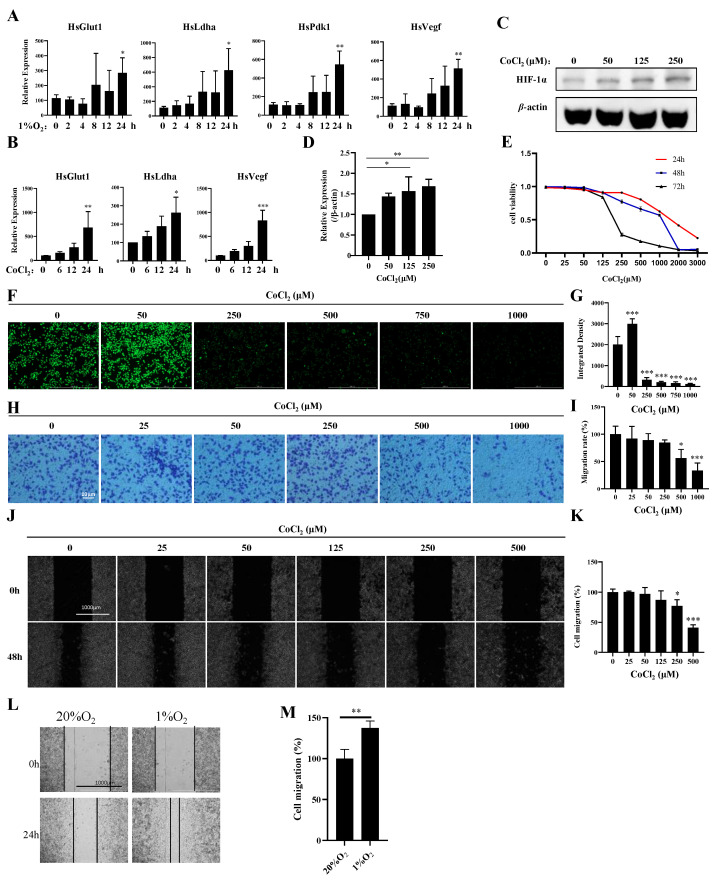
Establishment of a hypoxia model with 1% oxygen and CoCl_2_ in MDA-MB-231 cells. (**A**) MDA-MB-231 cells were treated with 1% O_2_ for the indicated times, and the mRNA levels of the indicated genes were analyzed by qPCR. (**B**) MDA-MB-231 cells were treated with 500 μM CoCl_2_ for the indicated times, and the mRNA levels of the indicated genes were analyzed by qPCR. (**C**,**D**) MDA-MB-231 cells were treated with the indicated concentrations of CoCl_2_ for 48 h, then the protein levels of the indicated proteins were analyzed by Western blotting. The quantification of the gray values of the indicated proteins from three independent experiments is shown. (**E**) MDA-MB-231 cells were treated with the indicated concentrations of CoCl_2_ for 48 h, and the cell viability was detected with a CCK-8 assay. (**F**,**G**) MDA-MB-231 cells were treated with the indicated concentrations of CoCl_2_ for 1 day and subjected to an ROS production assay. Scale bars: 1000 μm. (**H**,**I**) MDA-MB-231 cells were treated with the indicated concentrations of CoCl_2_ for 12 h and then subjected to an in vitro migration assay. The quantification of 1 × 10^5^ cells from three independent experiments is shown. Scale bars: 10 μm. (**J**,**K**) Wound healing of MDA-MB-231 cells in the presence of CoCl_2_. Scale bars: 1000 μm. Data information: Each value represents the mean ± SD. Statistical significance was evaluated by *t*-tests. * *p* < 0.05; ** *p* < 0.01; *** *p* < 0.001. The data show one representative of three independent experiments with three biological replicates. (**L**,**M**) Wound healing of MDA-MB-231 cells in the presence of 1% O_2_. Scale bars: 1000 μm. Data information: Each value represents the mean ± SD. Statistical significance was evaluated by *t*-tests. ** *p* < 0.01. The data show one representative of three independent experiments with three biological replicates.

**Figure 2 biology-13-00577-f002:**
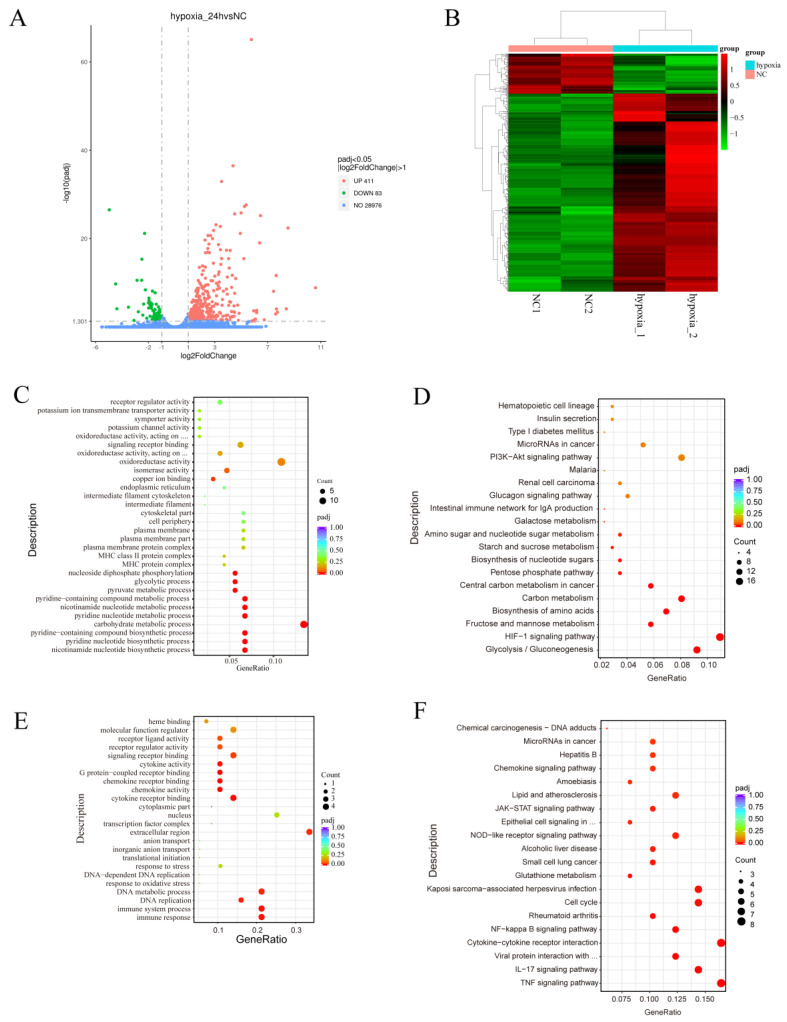
Functional annotation of the DEGs associated with the response of MDA-MB-231 cells to hypoxia. (**A**,**B**) Volcano map (**A**) and heatmap (**B**) of the DEGs after hypoxia. (**C**–**F**) GO annotation (**C**,**E**) and KEGG pathway enrichment (**D**,**F**) of the DEGs. NC: normoxia group.

**Figure 3 biology-13-00577-f003:**
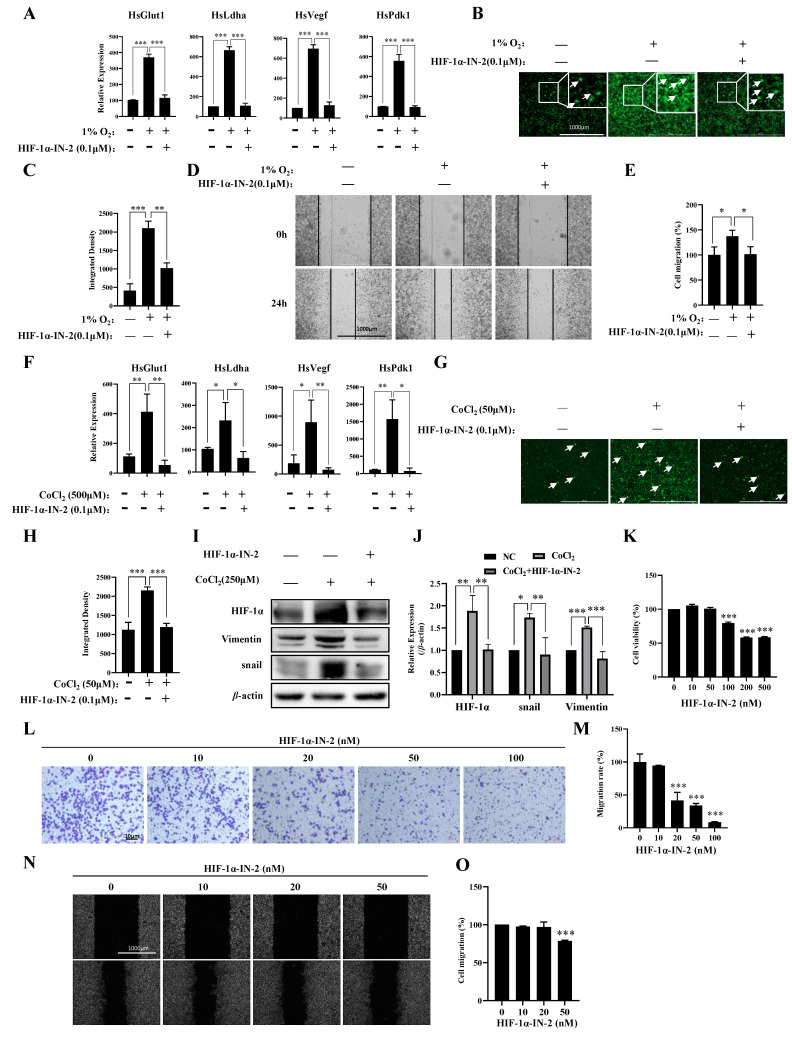
A HIF1α inhibitor inhibits multiple processes in MDA-MB-231 cells. (**A**) MDA-MB-231 cells were treated with 1% O_2_ in the presence or absence of HIF-1α-IN2 for the indicated times, and the mRNA levels of the indicated genes were analyzed by qPCR. (**B**,**C**) MDA-MB-231 cells were treated with 1% O_2_ in the presence or absence of HIF-1α-IN2 for the indicated times and subjected to an ROS production assay. Scale bars: 1000 μm. (**D**,**E**) MDA-MB-231 cells were treated with 1% O_2_ in the presence or absence of HIF-1α-IN2 for the indicated times and subjected to a wound-healing assay. Scale bars: 1000 μm. (**F**) MDA-MB-231 cells were treated with CoCl_2_ in the presence or absence of HIF-1α-IN2 for the indicated times, and the mRNA levels of the indicated genes were analyzed by qPCR. (**G**,**H**) MDA-MB-231 cells were treated with CoCl_2_ in the presence or absence of HIF-1α-IN2 for the indicated times and subjected to an ROS production assay. Scale bars: 1000 μm. (**I**,**J**) MDA-MB-231 cells were treated with CoCl_2_ in the presence or absence of HIF-1α-IN2 for the indicated times, and the protein levels of the indicated proteins were analyzed by Western blotting (Appendix A: uncropped gels). The quantification of the gray values of the indicated proteins from three independent experiments is shown. (**K**) Proliferation of MDA-MB-231 cells in the presence of HIF-1α-IN2 for 48 h. (**L**,**M**) MDA-MB-231 cells were treated with the indicated concentrations of HIF-1α-IN2 for 2 days and subjected to an in vitro migration assay. The quantification of cells at a density of 1 × 10^5^ per well from three independent experiments is shown. Scale bars: 10 μm. (**N**,**O**) MDA-MB-231 cells were treated with the indicated concentrations of HIF-1α-IN2 for two days and subjected to a wound healing assay. Scale bars: 1000 μm. Data information: Each value represents the mean ± SD. Statistical significance was evaluated by *t*-tests. * *p* < 0.05; ** *p* < 0.01; *** *p* < 0.001. The data represent one representative of three independent experiments with three biological replicates. The white arrows in (**B**,**G**) represent cells that produce ROS.

**Figure 4 biology-13-00577-f004:**
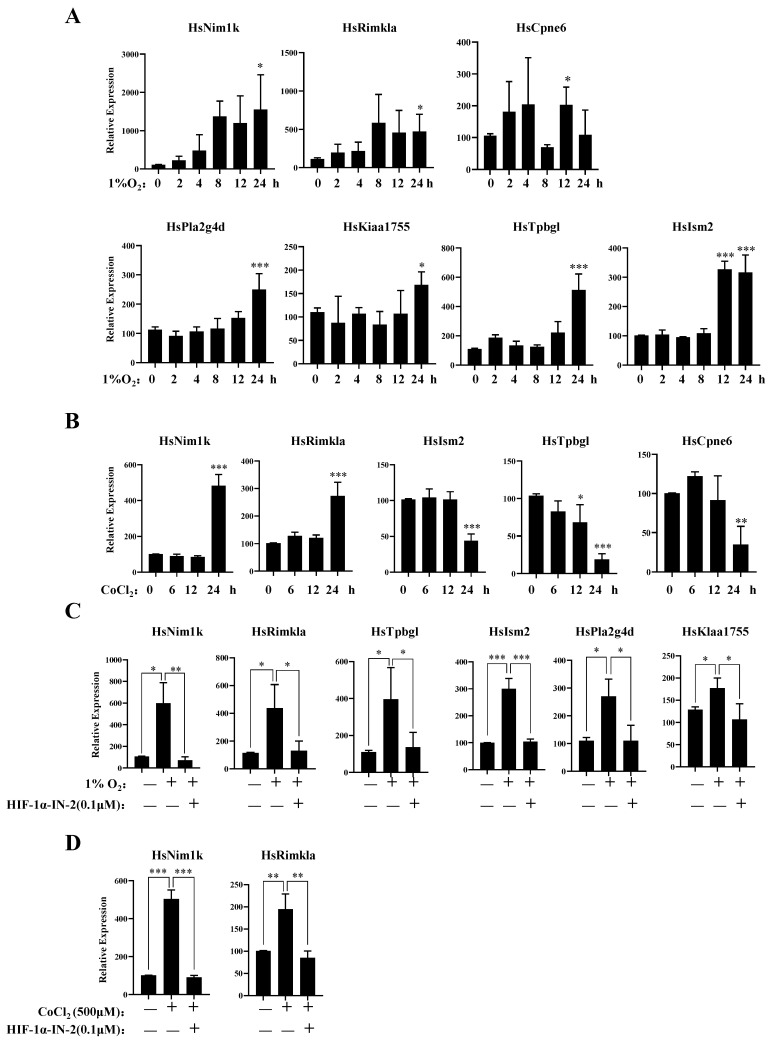
New gene response to hypoxia. (**A**) MDA-MB-231 cells were treated with 1% O_2_ for the indicated times, then the mRNA levels of the indicated genes were analyzed by qPCR. (**B**) MDA-MB-231 cells were treated with 500 μM CoCl_2_ for the indicated times, then the mRNA levels of the indicated genes were analyzed by qPCR. (**C**) MDA-MB-231 cells were treated with 1% O_2_ in the presence or absence of HIF-1α-IN2 for the indicated times, then the mRNA levels of the indicated genes were analyzed by qPCR. (**D**) MDA-MB-231 cells were treated with CoCl_2_ in the presence or absence of HIF-1α-IN2 for the indicated times, then the mRNA levels of the indicated genes were analyzed by qPCR. Data information: each value represents the mean ± SD. Statistical significance was evaluated by *t*-tests. * *p* < 0.05; ** *p* < 0.01; *** *p* < 0.001. The data represent one representative of three independent experiments with three biological replicates.

**Table 1 biology-13-00577-t001:** TFs that upregulate genes in response to hypoxia.

Key TF	# of Overlapped Genes	*p* Value	Q Value	List of Overlapped Genes
HIF1A	19	3.06 × 10^−16^	2.69 × 10^−14^	*LOX*, *CA9*, *VEGFA*, *FAM162A*, *PFKFB3*, *RORA*, *BNIP3*, *PFKFB4*, *PGK1*, *ITGB2*, *MUC1*, *EGLN3*, *GAPDH*, *ENO1*, *ALDOA*, *LDHA*, *SLC2A1*, *EGLN1*, *VEGFB*
SP1	19	0.000841	0.0185	*ITGB2*, *RORA*, *PPL*, *ITGA5*, *CDC42BPG*, *EPOR*, *NOS3*, *CA9*, *VEGFA*, *TGFB1*, *ITGB3*, *KRT18*, *LDHA*, *CDKN1A*, *TIMP3*, *KCTD11*, *NKX2-1*, *P4HA1*, *CAV1*
JUN	7	0.0174	0.0383	*SOX7*, *CDKN1A*, *PDK1*, *NOS3*, *NEFL*, *VEGFA*, *LDHA*
MYC	6	0.00895	0.0272	*VEGFA*, *NDRG1*, *LDHA*, *SFRP1*, *CDKN1A*, *PGK1*
SP3	6	0.0157	0.0363	*CA9*, *CDKN1A*, *RORA*, *NKX2-1*, *VEGFA*, *NOS3*
STAT3	6	0.0417	0.068	*VEGFA*, *VEGFB*, *TGFB1*, *CDKN1A*, *HSPB1*, *MUC1*
TP53	6	0.0733	0.104	*CAV1*, *VEGFA*, *CDKN1A*, *SLC2A1*, *DUSP1*, *PLAGL1*
PPARG	5	0.00639	0.0238	*CDKN1A*, *CAV1*, *ANGPTL4*, *PLIN2*, *VLDLR*
HDAC1	5	0.00866	0.0272	*NOS3*, *RUNX3*, *POU5F1*, *SFRP1*, *CDKN1A*
ETS1	5	0.0134	0.0327	*CDKN1A*, *ITGB3*, *NDRG1*, *TMEM158*, *CA9*
AR	5	0.0253	0.0473	*NDRG1*, *CDKN1A*, *MUC1*, *KISS1R*, *VEGFA*
ATM	4	0.00056	0.0164	*CDKN1A*, *DUSP1*, *SLC2A1*, *VEGFA*
DNMT1	4	0.00212	0.0191	*TIMP3*, *RUNX3*, *SFRP1*, *VEGFA*
SMAD3	4	0.00212	0.0191	*VEGFA*, *ANGPTL4*, *TGFB1*, *CDKN1A*
EZH2	4	0.00544	0.0238	*CDKN1A*, *RUNX3*, *ADAMTS1*, *SFRP1*
VDR	4	0.00648	0.0238	*BHLHE40*, *DDIT4*, *CDKN1A*, *HLA-DRB1*
MYCN	4	0.00828	0.0272	*DKK3*, *NDRG1*, *CDKN1A*, *MXI1*
E2F1	4	0.217	0.244	*CDKN1A*, *DUSP1*, *VEGFA*, *ISYNA1*
NFKB1	4	0.791	0.8	*CDKN1A*, *VEGFA*, *TGFB1*, *NOS3*

HIF1A: hypoxia inducible factor 1, alpha subunit (basic helix-loop-helix transcription factor); SP1: Sp1 transcription factor; JUN: jun proto-oncogene; MYC: v-myc myelocytomatosis viral oncogene homolog (avian); SP3: Sp3 transcription factor; STAT3: signal transducer and activator of transcription 3 (acute-phase response factor); TP53: tumor protein p53; PPARG: peroxisome proliferator-activated receptor gamma; HDAC1: histone deacetylase 1; ETS1: v-ets erythroblastosis virus E26 oncogene homolog 1 (avian); AR: androgen receptor; ATM: ataxia telangiectasia mutated; DNMT1: DNA (cytosine-5-)-methyltransferase 1; SMAD3: SMAD family member 3; EZH2: enhancer of the zeste homolog 2; VDR: (Drosophila) vitamin D (1,25-dihydroxyvitamin D3) receptor; MYCN: v-myc myelocytomatosis virus-related oncogene, neuroblastoma-derived (avian); E2F1: E2F transcription factor 1; NFKB1: nuclear factor of kappa light polypeptide gene enhancer in B cells 1.

**Table 2 biology-13-00577-t002:** TFs that downregulate genes in response to hypoxia.

Key TF	# of Overlapped Genes	*p*-Value	Q Value	List of Overlapped Genes
SP1	7	0.00413	0.00744	*CYP1B1*, *GDA*, *AREG*, *IL6*, *PTGS2*, *CXCL1*, *ODC1*
E2F1	6	0.000024	0.000216	*RRM2*, *CDC6*, *CDCA7*, *MEFV*, *UHRF1*, *PCNA*
RELA	6	0.00186	0.00408	*IL6*, *PTGS2*, *CCL20*, *CXCL2*, *OLR1*, *CXCL1*
NFKB1	6	0.00193	0.00408	*CCL20*, *PTGS2*, *IL6*, *CXCL1*, *CXCL2*, *OLR1*
JUND	4	1.32 × 10^−5^	0.000159	*IL6*, *NQO1*, *GADD45A*, *PTGS2*
EP300	4	9.76 × 10^−5^	0.000442	*PTGS2*, *CYP1B1*, *PCNA*, *IL6*
BRCA1	4	0.000105	0.000442	*CYP1B1*, *GADD45A*, *AREG*, *CXCL1*
CREB1	4	0.000608	0.00199	*SLC20A1*, *ODC1*, *IL6*, *PTGS2*
MYC	4	0.000903	0.0027	*CDCA7*, *IL6*, *PCNA*, *ODC1*

SP1: Sp1 transcription factor; E2F1: E2F transcription factor 1; RELA: v-rel reticuloendotheliosis viral oncogene homolog A (avian); NFKB1: nuclear factor of kappa light polypeptide gene enhancer in B cells 1; JUND: jun D proto-oncogene; EP300: E1A binding protein p300; BRCA1: breast cancer 1, early onset; CREB1: cAMP responsive element binding protein 1; MYC: v-myc myelocytomatosis viral oncogene homolog (avian).

## Data Availability

All data generated or analyzed during this study are included in this published article (and its Appendix A).

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
