# Peer review of "Targeting Hypoxia and HIF1α in Triple-Negative Breast Cancer: New Insights from Gene Expression Profiling and Implications for Therapy"

_biology, 2024, doi:10.3390/biology13080577_

Round 1
Reviewer 1 Report
Comments and Suggestions for Authors
The article “Hypoxia Induces a Significant Change in Gene Expression in MDA-MB-231 Cells” describes the effect of either hypoxia or CoCl2 on the triple-negative breast cancer cell line MDA-MB-231 defined by measuring gene expression modulation, ROS production and invasive potential.
The authors showed that CoCl2 only partially mimics the effects of low oxygen-induced hypoxia. CoCl2 induces a small increase in HIF-1a protein levels, the gene expression upregulation of a subset of HIF-1a target genes, but an inhibition of the migratory potential of MDA-MB-231. Interestingly, CoCl2 induces ROS production at low concentrations, while it inhibits it at higher concentrations. The authors demonstrate a profound effect of hypoxia on the gene expression landscape of MDA-MB-231, which leads to the upregulation of a wide subset of genes, mostly already known targets of the transcription factor HIF-1a, and the downregulation of a smaller subset of genes.
They utilised the HIF-1a inhibitor HIF-1a-1N-2 to decrease gene expression of HIF-1a target genes, ROS production and migratory potential induced by oxygen depletion and CoCl2. Since MDA-MB-231 cell line is characterised by high levels of HIF-1a, the authors demonstrated that the inhibitor is capable of decreasing the invasive potential of the cells even in normoxic conditions.
Finally, the authors evidence the upregulation of a group of genes which they identify as “new responders” to hypoxia, which they (partially) validate with time courses of hypoxia, CoCl2 and by means of the inhibitor HIF-1a-1N-2.
This article has the potential of providing to the scientific community an accurate description of the biological changes induced by hypoxia and the effects of a hypoxia inducer and an inhibitor in a model of triple-negative breast cancer. However, in the actual state it fails in accuracy and significance. Therefore, major revisions are required. The major points to be addressed are listed below:
- CoCl2 has not been described in sufficient detail and put in the context of the scientific literature. Its partial effect of hypoxia-like response, which fails to induce migration has been widely described in the literature, therefore its role on MDA-MB-231 needs to be accurately put in the context of triple negative breast cancer.
An effect of CoCl2 on cell viability needs to be demonstrated, which can potentially explain its activation of ROS production at low concentration and its inhibition at high concentration.
- RNA-sequencing experiments need to be fully described in the material and methods section, and the raw data need to be deposited in GEO repositories. A careful description of the methods used to induce hypoxia in the samples used for RNA-seq is missing.
The gene expression modulation induced by hypoxia has been widely demonstrated, therefore the value of this set of results is incremental. This set of data would acquire a high relevance if compared with the RNA-seq data of MDA-MB-231 treated with CoCl2.
- The new genes listed as responding to hypoxia need to be described and put in a potential mechanistic context, to the best of the possibilities, otherwise it results in a list of genes name without potential interest. The analysis of the set of genes need to be complete for each treatment.
- The manuscript requires extensive editing throughout, starting from the title. Changing title is recommended, in order to provide a more careful description of the data demonstrated.
Comments on the Quality of English LanguageA full editing is necessary. A numerous amount of typos is detectable throughout the manuscript, especially in the discussion section.
I recommend to change the title, to be more representative of the results described in the manuscript.
Author Response
Overall Response: We thank all reviewers for their valuable time and insightful comments on our manuscript. We appreciate the constructive feedback you have provided, which has helped us to improve the quality of our work. In this point-by-point response letter, reviewers’ comments were marked in dark blue italics, followed by our point-by-point responses. All revised or supplementary new text is underlined in red.
Reviewer #1:
The article “Hypoxia Induces a Significant Change in Gene Expression in MDA-MB-231 Cells” describes the effect of either hypoxia or CoCl2 on the triple-negative breast cancer cell line MDA-MB-231 defined by measuring gene expression modulation, ROS production and invasive potential.
The authors showed that CoCl2 only partially mimics the effects of low oxygen-induced hypoxia. CoCl2 induces a small increase in HIF1a protein levels, the gene expression upregulation of a subset of HIF1a target genes, but an inhibition of the migratory potential of MDA-MB-231. Interestingly, CoCl2 induces ROS production at low concentrations, while it inhibits it at higher concentrations. The authors demonstrate a profound effect of hypoxia on the gene expression landscape of MDA-MB-231, which leads to the upregulation of a wide subset of genes, mostly already known targets of the transcription factor HIF-1a, and the downregulation of a smaller subset of genes.
They utilised the HIF-1a inhibitor HIF-1a-1N-2 to decrease gene expression of HIF-1a target genes, ROS production and migratory potential induced by oxygen depletion and CoCl2. Since MDA-MB-231 cell line is characterised by high levels of HIF-1a, the authors demonstrated that the inhibitor is capable of decreasing the invasive potential of the cells even in normoxic conditions.
Finally, the authors evidence the upregulation of a group of genes which they identify as “new responders” to hypoxia, which they (partially) validate with time courses of hypoxia, CoCl2 and by means of the inhibitor HIF-1a-1N-2.
This article has the potential of providing to the scientific community an accurate description of the biological changes induced by hypoxia and the effects of a hypoxia inducer and an inhibitor in a model of triple-negative breast cancer. However, in the actual state it fails in accuracy and significance. Therefore, major revisions are required. The major points to be addressed are listed below:
- 1. CoCl2 has not been described in sufficient detail and put in the context of the scientific literature. Its partial effect of hypoxia-like response, which fails to induce migration has been widely described in the literature, therefore its role on MDA-MB-231 needs to be accurately put in the context of triple negative breast cancer.
Response:
Thank you for your insightful feedback regarding the description of CoCl2. We acknowledge the need for a more comprehensive and contextually relevant discussion of its role in the study. CoCl2 is a widely used hypoxia mimetic that can be a valuable tool for studying the role of hypoxia in TNBC. However, it is important to be aware of the limitations of CoCl2 and to interpret results obtained using this compound with caution. Further research is needed to fully understand the complex relationship between hypoxia and TNBC progression. We have added the detail description of CoCl2 and its’ role in TNBC on page 14 line 387 to line 400.
page 14 line 387 to line 400
CoCl2 is a chemical compound that has been widely used as a hypoxia mimetic in research studies[1]. CoCl2 mimics the effects of hypoxia by stabilizing hypoxia-inducible factor, which plays a critical role in the cellular response to low oxygen levels. Hypoxia has been implicated in the development and progression of TNBC. Hypoxic condition promotes tumor growth, angiogenesis, and metastasis. Additionally, hypoxia contribute to resistance to chemotherapy and radiotherapy. CoCl2 can be a valuable tool for studying the role of hypoxia in TNBC. By treating TNBC cells with CoCl2, researchers can investigate the effects of hypoxia on cell proliferation, survival, migration, and invasion[2]. These studies can provide insights into the mechanisms by which hypoxia contributes to TNBC progression and identify potential therapeutic targets. However, CoCl2 has been shown to have effects that are independent of HIF, which can complicate the interpretation of results obtained using this compound. CoCl2 were reported to inhibit the proliferation of TNBC cells, and the combined treatment of glibenclamide and CoCl2 inhibits growth in highly metastatic breast cancer[3].
- 2. An effect of CoCl2 on cell viability needs to be demonstrated, which can potentially explain its activation of ROS production at low concentration and its inhibition at high concentration.
Response:
Thank you for your valuable feedback. We appreciate your suggestion to investigate the effect of CoCl2 on cell viability. We have conducted a CCK-8 assay to assess cell viability in response to different concentrations of CoCl2. Our results demonstrate that high concentrations of CoCl2 do indeed lead to a decrease in cell viability. This finding supports our hypothesis that the contrasting effects of CoCl2 on ROS production at different concentrations may be related to its impact on cell viability.
Specifically, at low concentrations, CoCl2 have a minimal impact on cell viability, allowing cells to respond to the hypoxia-like conditions by activating ROS production as a defense mechanism. However, at high concentrations, CoCl2 may become toxic to cells, leading to decreased cell viability and ultimately inhibiting ROS production. We have now compiled the data into revised Figure 1E and described them in revised text from page 4, line 210.
page 4, line 210.
Furthermore, we found that CoCl2 treatment could activate HIF1α (Figure 1C, 1D), inhibit proliferation (Figure 1E), and induce ROS production and apoptosis (Figure 1F, 1G; Figure S1A).
- 3. RNA-sequencing experiments need to be fully described in the material and methods section, and the raw data need to be deposited in GEO repositories. A careful description of the methods used to induce hypoxia in the samples used for RNA-seq is missing.
Response:
Thank you for your valuable feedback. We appreciate your suggestion to provide a more detailed description of our RNA-sequencing experiments and to make the raw data publicly available. We have revised the materials and methods section of our manuscript to include a comprehensive description of the RNA-sequencing methods used. We have also deposited the raw data from our RNA-sequencing experiments in the GEO database. The GEO accession number (PRJNA1132955) is now included in the revised materials and methods section of our manuscript. We have added a description of the methods used to the materials and methods section of our revised manuscript on page 6, line 172 to line 201.
page 6, line 172 to line 201
MDA-MB-231 cells under 1% O2 for hypoxia or 21% O2 for normoxia for 24 h were collected. The methods used for total RNA extraction and RNA-seq were the same as previously described[4]; briefly, RNA integrity was assessed using an RNA Nano 6000 Assay Kit for the Bioanalyzer 2100 system (Agilent Technologies, CA, USA)[5]. Then, the mRNA was purified, and first-strand cDNA was synthesized. Second-strand cDNA synthesis was subsequently performed using DNA Polymerase I and RNase H[6]. After adenylation of the 3’ ends of the DNA fragments, adaptors with hairpin loop structures were ligated to prepare for hybridization. The library fragments were purified with the AMPure XP system[7]. Then, PCR was performed with Phusion High-Fidelity DNA polymerase, universal PCR primers and index (X) primers. Finally, the PCR products were purified (AMPure XP system), and library quality was assessed on an Agilent Bioanalyzer 2100 system. Clustering of the index-coded samples was performed on a c Bot Cluster Generation System using the TruSeq PE Cluster Kit v3-cBot-HS (Illumina) according to the manufacturer’s instructions[8]. After cluster generation, the library preparations were sequenced on an Illumina Nova seq platform, and 150 bp paired-end reads were generated.
The raw data (raw reads) in fastq format were first processed through fastp software. In this step, clean data (clean reads) were obtained by removing reads containing adapters, reads containing poly-N sequences and low-quality reads from the raw data. The index of the reference genome was built using HISAT2 v2.0.5, and paired-end clean reads were aligned to the reference genome using HISAT2 v2.0.5[9]. The mapped reads of each sample were assembled by String Tie (v1.3.3b) in a reference-based approach[10]. Feature Counts v1.5.0-p3 was used to count the number of reads mapped to each gene[11]. The FPKM value of each gene was subsequently calculated based on the length of the gene and the number of reads mapped to that gene. Differential expression analysis and Gene Ontology (GO) and Kyoto Encyclopedia of Genes and Genomes (KEGG) enrichment analyses of the DEGs were implemented in the cluster Profiler R package, in which gene length bias was corrected[12] [13, 14]. GO terms with corrected P values less than 0.05 were considered significantly enriched in the DEGs. We have also deposited the raw data from our RNA-sequencing experiments in the GEO database (PRJNA1132955).
- 4. The gene expression modulation induced by hypoxia has been widely demonstrated, therefore the value of this set of results is incremental. This set of data would acquire a high relevance if compared with the RNA-seq data of MDA-MB-231 treated with CoCl2.
Response:
Thank you for your valuable feedback. We appreciate your suggestion to compare our results with RNA-sequencing data from MDA-MB-231 cells treated with CoCl2. We acknowledge that this comparison would provide additional insights into the effects of CoCl2 on gene expression in MDA-MB-231 cells. However, due to time constraints, we were unable to conduct these additional experiments for this study. We plan to investigate this aspect further in future research. We have included a statement in our revised manuscript acknowledging the limitations of the current study, outlining our plans for future investigations, and described them in revised text from page 14, line 408 to line 416.
page 14, line 408 to line 416
Our study did not perform RNA-sequencing analysis on CoCl2-treated cells compared with cells exposed to actual hypoxia. This comparison would have provided valuable insights into the similarities and differences between CoCl2-induced and hypoxia-induced gene expression changes in MDA-MB-231 cells. Future studies should aim to address this limitation by conducting RNA-sequencing analysis on both CoCl2-treated and hypoxic MDA-MB-231 cells. This study provides a more comprehensive understanding of the effects of CoCl2 on gene expression and its potential as a tool for studying hypoxia-related mechanisms in triple-negative breast cancer research.
- 5. The new genes listed as responding to hypoxia need to be described and put in a potential mechanistic context, to the best of the possibilities, otherwise it results in a list of genes name without potential interest. The analysis of the set of genes need to be complete for each treatment.
Response:
Thank you for your careful review and insightful comments. We have included a description of the new genes in our revised manuscript from page 11, line 337 to line 357.
page 11, line 337 to line 357
Nim1k is a serine/threonine protein kinase, and serine/threonine protein kinases play important roles in signal transmission under hypoxia, such as the PI3K/AKT signal transduction pathway[15] and the mTOR signal transduction pathway[16]. Thus, Nim1k may participate in signal transmission under hypoxia. Rimkla is a ribosomal modification protein rimK-like family member A that is a NAAG synthetase (NAAGS-II), and NAAG protects against injury induced by NMDA and hypoxia[17, 18]. Cpne6 is a calcium-binding protein that can trigger calcium signals by activating and recruiting the Rho GTPase Rac1 to cell membranes[19]. In breast cancer cells, the regulation of [Ca2+]i is crucial for tumorigenesis and the development of cancer hallmarks, including cell growth and proliferation, migration, metastasis and resistance to apoptosis[20, 21]. Cpne6 may play important roles in regulating Ca2+ influx. Tpbgl is predicted to be involved in the negative regulation of the canonical Wnt signaling pathway. Research has indicated that high expression of Kiaa1755 is present in breast cancer patients[22], but our results revealed that the expression of Kiaa1755 is upregulated in MDA-MB-231 cells under hypoxia, which should be further studied in the future. Pla2g4d belongs to the phospholipase A2 (PLA2) enzyme family, which catalyzes the hydrolysis of glycerophospholipids at the sn-2 position and then liberates arachidonic acid (AA) and lysophospholipids[23]. PLA2 is involved in multiple diseases, such as cardiovascular disease[24], Parkinson's disease[25], and Alzheimer's disease[26]. High expression of PLA2 indicates the malignant potential of human breast cancer [27, 28]. The function of Ism2 is not clear, and the role of Ism2 in MDA-MB-231 cells remains to be further studied.
6、The manuscript requires extensive editing throughout, starting from the title. Changing title is recommended, in order to provide a more careful description of the data demonstrated.
Response:
Thank you for your thorough review and valuable feedback. We acknowledge the need for extensive editing throughout the manuscript, including the title. We have carefully considered your suggestions and have made significant revisions to improve the clarity, accuracy, and overall quality of the revised manuscript.
Comments on the Quality of English Language
1、A full editing is necessary. A numerous amount of typos is detectable throughout the manuscript, especially in the discussion section.
Response:
Thank you for pointing out the typos in the manuscript. We apologize for these errors and are committed to ensuring that the final version of the manuscript is free of any such mistakes. We have conducted a thorough review of the manuscript and have corrected all of the identified typos in the revised manuscript.
2、I recommend to change the title, to be more representative of the results described in the manuscript.
Response:
Thank you for your suggestion to change the title of our manuscript. We appreciate your feedback and have carefully considered your recommendation. We agree that the original title, " Hypoxia Induces a Significant Change in Gene Expression in MDA-MB-231 Cells" did not fully capture the breadth and depth of our findings. Therefore, we have revised the title to: “Targeting Hypoxia and HIF1α in Triple-Negative Breast Cancer: New Insights from Gene Expression Profiling and Implications for Therapy” on page 1 line 1.
on page 1 line 1
Targeting Hypoxia and HIF1α in Triple-Negative Breast Cancer: New Insights from Gene Expression Profiling and Implications for Therapy.
Reviewer 2 Report
Comments and Suggestions for Authors
The authors demonstrated HIF1a signaling pathway and targeted genes can be used as therapeutic target in breast cancer. And they also found the effect of CoCl2-induced hypoxia is different with low oxygen condition. From RNA sequencing results, some new HIF1a targeted genes were identified and validated. But there are some major comments.
1. Please provide the rationale of the CoCl2 concentration used to mimic hypoxia. Previous paper discussed the effect of CoCl2 causing cell apoptosis and inhibiting cell proliferation. Is the inhibition of cell migration caused by inhibition of cell growth or induction of cell apoptosis?
2. Please discuss how HIF-1α-IN-2 can be used in clinic. Is there any animal experiment done using HIF-1α-IN-2?
3. Please clarify the hypoxia method. And what is the hypoxia condition used when you do the RNAseq?
4. In your supplemental file, only one time WB results were shown. Please provide all three repeats and label the molecular weight of protein ladders.
And there are some minor comments:
1. In introduction session, please provide more detail information of current breast cancer therapies and how resistance occur (current drugs and its targets).
2. Figure 1C, please provide the quantification results from three WB results.
3. Line 218, typo Vimentin
Comments on the Quality of English Language
Need to improve
Author Response
Overall Response: We thank all reviewers for their valuable time and insightful comments on our manuscript. We appreciate the constructive feedback you have provided, which has helped us to improve the quality of our work. In this point-by-point response letter, reviewers’ comments were marked in dark blue italics, followed by our point-by-point responses. All revised or supplementary new text is underlined in red.
The authors demonstrated HIF1a signaling pathway and targeted genes can be used as therapeutic target in breast cancer. And they also found the effect of CoCl2-induced hypoxia is different with low oxygen condition. From RNA sequencing results, some new HIF1a targeted genes were identified and validated. But there are some major comments.
- 1. Please provide the rationale of the CoCl2 concentration used to mimic hypoxia. Previous paper discussed the effect of CoCl2 causing cell apoptosis and inhibiting cell proliferation. Is the inhibition of cell migration caused by inhibition of cell growth or induction of cell apoptosis?
Response:
Thank you for your question about the rationale for the CoCl2 concentration used in our study and its relationship to cell migration inhibition. We appreciate your interest in our research and are happy to provide additional information. The choice of CoCl2 concentration depends on several factors, including cell type, experimental goals, duration of treatment, and previous studies. In our study, we used different concentrations of CoCl2 (0.05 µM - 6 µM) to determine the dose-dependent effects of CoCl2 in MDA-MB-231 cells, as evidenced by the accumulation of HIF1α (Figure 1C).
Mechanism of migration inhibition:
Based on our findings, we found that the inhibition of cell migration observed in our study is primarily due to inhibit the cell proliferation, rather than an indirect effect mediated by cell apoptosis. We observed significant growth inhibition at the concentration of CoCl2 used (0.5 µM for 48 hours) (Revised Figure 1E). However, we did not observe significant cell apoptosis at the concentration of CoCl2 used (0.5 µM for 24 hours) (Revised Figure S1A). We have now included the cell proliferation results in revised Figure 1E and the apoptosis results in revised Figure S1A. We have also added the description of these results on page 5 line 210.
on page 5 line 210
Furthermore, we found that CoCl2 treatment could activate HIF1α (Figure 1C, 1D), inhibit proliferation (Figure 1E), and induce ROS production and apoptosis (Figure 1F, 1G; Figure S1A)
2.Please discuss how HIF-1α-IN-2 can be used in clinic. Is there any animal experiment done using HIF-1α-IN-2?
Response:
Thank you for your question about the clinical applications of HIF-1α-IN-2 and the plans for future animal experiments. We appreciate your interest in our research and are happy to provide additional information. We have clarified some potential clinical applications of HIF-1α-IN-2 in the discussion sections of revised manuscript from page 16, line 474 to line 486. We agree that animal experiments are an important next step in our research. We plan to conduct animal experiments in future studies to further investigate the efficacy and safety of HIF-1α-IN-2 in vivo. These studies will allow us to evaluate the effects of HIF-1α-IN-2 on tumor growth, metastasis, and overall survival in animal models of TNBC and other cancers.
page 16, line 474 to line 486
Our results revealed that a HIF1α inhibitor could efficiently inhibit ROS production; inhibit the proliferation and migration of MDA-MB-231 cells; reduce the mRNA expression of the angiogenesis-associated gene HsVegf and the glycolysis-associated genes HsLdha, HsPdk1, and HsGlut1; and decrease the protein levels of the EMT-associated proteins SNAIL and VIMENTIN. These results suggest that HIF-1α-IN-2, a potent and selective inhibitor of HIF-1α, has promising potential for clinical applications in the treatment of TNBC. HIF-1α-IN-2 could be used in combination with radiotherapy and chemotherapy to enhance their efficacy and overcome resistance. In addition, HIF-1α is overexpressed in many other types of cancer, including clear cell renal cell carcinoma, head and neck cancer, and pancreatic cancer. Moreover, HIF-1α-IN-2 could be used to treat these cancers. Thus, HIF-1α-IN-2 is a promising new therapeutic agent with potential applications in the treatment of TNBC and other cancers. Further research is needed to fully evaluate its clinical potential and to develop optimal treatment strategies.
3.Please clarify the hypoxia method. And what is the hypoxia condition used when you do the RNA-seq?
Response:
Thank you for your careful review and insightful comments. We appreciate you bringing the missing information on hypoxia preparation methods to our attention. We have added a description of the methods used to prepare hypoxic cells to the materials and methods section of our revised manuscript on page 3, line 113, and page 5, line 173.
page 3, line 113.
under 1% O2 for hypoxia or 21% O2 for normoxia experiments.
page 5, line 173.
MDA-MB-231 cells under 1% O2 for hypoxia or 21% O2 for normoxia 24 h were collected.
4.In your supplemental file, only one time WB results were shown. Please provide all three repeats and label the molecular weight of protein ladders.
Response:
Thank you for your careful review and for highlighting the importance of providing complete and accurate information in the supplementary materials. We appreciate you bringing the missing information about the western blot replicates and protein ladder labels to our attention. We have added the remaining two western blot replicates to the revised supplementary file and have also labeled the molecular weight of the protein ladders in all three blots. Thank you for your careful review and for bringing the misidentification of the HIF1α band in Figure 1C to our attention. We sincerely apologize for this error and for any confusion it may have caused. We have carefully reviewed the western blot data and have identified the correct HIF1α band, which is located between 100 kDa and 130 kDa. We have corrected Figure 1C to show the correct HIF1α band in the revised Figure 1C and have also provided an image of the band in the revised supplementary materials, highlighted with a red box. We are grateful for your feedback and for helping us to improve the accuracy of our study. We have taken steps to ensure that such errors do not occur in the future.
And there are some minor comments:
1.In introduction session, please provide more detail information of current breast cancer therapies and how resistance occur (current drugs and its targets).
Response:
Thank you for your valuable feedback. We appreciate your suggestion to provide more detailed information on current breast cancer therapies and resistance mechanisms in the introduction section. We have added a description of the detail information into the introduction section on page 2, line 60 to line 80.
page 2, line 60 to line 80
The optimal treatment strategy for breast cancer depends on the tumor subtype, stage, and individual patient characteristics[29]. Prolonging life and palliating symptoms are the main goals for patients with metastatic breast cancer[29]. For nonmetastatic breast cancer, endocrine therapy is used mainly for patients with hormone receptor–positive tumors, and a minority of patients also receive chemotherapy[30]; ERBB2-targeted antibody or small-molecule inhibitor therapy combined with chemotherapy is used mainly for patients with ERBB2-positive tumors [31]; patients with triple-negative tumors receive chemotherapy alone[31]. Local therapy for all patients with nonmetastatic breast cancer consists of surgical resection, with consideration of postoperative radiation if lumpectomy is performed[32]. Additionally, targeted therapies, particularly those targeting epidermal growth factor receptor (EGFR), poly (ADP-ribose) polymerase 1(PARP1), vascular endothelial growth factor (VEGF), cyclin-dependent kinases 4 and 6(CDK4/6), and phosphatidylinositol-3-kinase (PI3K), have significantly improved the survival rates of patients with breast cancer[33]. However, targeted therapies are prone to developing resistance, leading to treatment failure. Resistance can be categorized into intrinsic and acquired resistance. Intrinsic resistance includes genetic mutations, activation of defense pathways, and cancer stem cell activation. Acquired resistance mechanisms include oncogene activation, changes to the tumor microenvironment[34], epigenetic modifications, enhanced DNA damage repair, and gene expression changes due to mutations[35]. For a detailed review of drug resistance mechanisms, please refer to the comprehensive article by Rajesh’s group[35].
2.Figure 1C, please provide the quantification results from three WB results.
Response:
Thank you for your careful review and insightful comments. We appreciate you bringing the request for quantification results to our attention. We have added the quantification results from the three western blot experiments to revised Figure 1D, and described them in revised text from page 6, line 223 to line 224.
page 6, line 223 to line 224.
The quantification of the gray values of the indicated proteins from three independent experiments is shown.
- 3. Line 218, typo Vimentin
Response:
Thank you for your careful review and insightful comments. We appreciate you bringing the spelling error to our attention. We have reviewed the manuscript and corrected the spelling of "Vimentin" on line 218. The word is now spelled correctly as "Vimentin" on page 9, line 277.
Reviewer 3 Report
Comments and Suggestions for Authors
The Ms entitled “Hypoxia Induces a Significant Change in Gene Expression in 2 MDA-MB-231 Cells” by Han et al. describes the changes, in terms of ion channel expression, induced by hypoxia and the hypoxia-mimicking compound CoCl2. Using a mRNA-seq technology they found that numerous genes involved in cell proliferation, migration and tumor resistance are upregulated, while others involved in the immune response are downregulated. Interestingly, CoCl2 appears not fully mimic the effect of hypoxia.
Given the relevance of the hypoxic environment in determining solid tumor aggressiveness, the Ms appears interesting. However, several points need to be clarified in order for the Ms to be published.
- Materials and Methods. No information on the methods used to prepare hypoxic cells is given.
- In Figure 1 the authors should also show the effects of 1% O2 on the transwell and wound-healing migration. Even if these experiments are present in the literature, it is important to show the difference with CoCl2, both conditions tested by the same laboratory. The effect of CoCl2 on the wound-healing and transwell migration appears at very high concentrations. The authors should discuss this.
- Figure 2. Similar experiments should be done with CoCl2, so that the differences in the up- and downregulated genes can be appreciated.
- Figure 3. The effect of the HIF-1alfa inhibitor on the wound-healing appears at too high concentration to be considered to be mediated by HIF.
- In the introduction and Discussion, it should be clarified that not all effects of hypoxia in tumors are mediated by HIF (see for example PMID: 38136613; PMID: 27028592; PMID: 25642170).
Comments on the Quality of English LanguageThe English needs to be significantly improved.
Author Response
Overall Response: We thank all reviewers for their valuable time and insightful comments on our manuscript. We appreciate the constructive feedback you have provided, which has helped us to improve the quality of our work. In this point-by-point response letter, reviewers’ comments were marked in dark blue italics, followed by our point-by-point responses. All revised or supplementary new text is underlined in red.
The Ms entitled “Hypoxia Induces a Significant Change in Gene Expression in 2 MDA-MB-231 Cells” by Han et al. describes the changes, in terms of ion channel expression, induced by hypoxia and the hypoxia-mimicking compound CoCl2. Using a mRNA-seq technology they found that numerous genes involved in cell proliferation, migration and tumor resistance are upregulated, while others involved in the immune response are downregulated. Interestingly, CoCl2 appears not fully mimic the effect of hypoxia.
Given the relevance of the hypoxic environment in determining solid tumor aggressiveness, the Ms appears interesting. However, several points need to be clarified in order for the Ms to be published.
1.Materials and Methods. No information on the methods used to prepare hypoxic cells is given.
Response:
Thank you for your careful review and insightful comments. We appreciate you bringing the missing information on hypoxia preparation methods to our attention. We have added a description of the methods used to prepare hypoxic cells to the materials and methods section of our revised manuscript on page 3, line 113.
on page 3, line 113.
under 1% O2 for hypoxia or 21% O2 for normoxia experiments.
- In Figure 1 the authors should also show the effects of 1% O2 on the transwell and wound-healing migration. Even if these experiments are present in the literature, it is important to show the difference with CoCl2, both conditions tested by the same laboratory. The effect of CoCl2 on the wound-healing and transwell migration appears at very high concentrations. The authors should discuss this.
Response:
Thank you for your insightful comment and for suggesting the inclusion of additional data on the effects of 1% O2 on transwell and wound-healing migration in Figure 1. We agree that it is important to show the difference between the effects of 1% O2 and CoCl2, both conditions tested by the same laboratory. We have added a new panel to Figure 1 showing the results of wound-healing migration assays performed under 1% O2 conditions. Compared to nomoxia, 1% O2 can promote the migration of MDA-MB-231 cells (Revised Figure 1L, 1M). We have added the data to revised Figure 1, and described them in revised text from page 5, line 211 to line 213.
In our study, we used different concentrations of CoCl2 (0.05 µM - 6 µM) to determine the dose-dependent effects of CoCl2 in MDA-MB-231 cells, as evidenced by the accumulation of HIF1α (Figure 1C).
Mechanism of Migration Inhibition:
Based on our findings, we found that the inhibition of cell migration observed in our study is primarily due to inhibit the cell proliferation, rather than an indirect effect mediated by cell apoptosis. We observed significant growth inhibition at the concentration of CoCl2 used (0.5 µM for 48 hours) (Revised Figure 1E). However, we did not observe significant cell apoptosis at the concentration of CoCl2 used (0.5 µM for 24 hours) (Revised Figure S1A). We have now included the cell proliferation results in revised Figure 1E and the apoptosis results in revised Figure S1A. We have also added the description of these results on page 5 line 210.
page 5, line 211 to line 213
However, CoCl2 treatment inhibited invasion and migration, which contradicts the high invasive and migration of MDA-MB-231 cells under hypoxia (Figure 1H-1M).
on page 5 line 210
Furthermore, we found that CoCl2 treatment could activate HIF1α (Figure 1C, 1D), inhibit proliferation (Figure 1E), and induce ROS production and apoptosis (Figure 1F, 1G; Figure S1A).
- 3. Figure 2. Similar experiments should be done with CoCl2, so that the differences in the up- and downregulated genes can be appreciated.
Response:
Thank you for your insightful comment and for suggesting the inclusion of additional data on the effects of CoCl2 on gene expression in Figure 2. We acknowledge that it would be valuable to compare the effects of CoCl2 and 1% O2 on gene expression to better understand the differences in their mechanisms of action. However, due to time constraints, we were unable to perform these additional experiments. We plan to investigate this aspect further in future research. We have included a statement in our revised manuscript acknowledging the limitations of the current study, outlining our plans for future investigations, and described them in revised text from page 14, line 408 to line 416.
page 14, line 408 to line 416
Our study did not perform RNA-sequencing analysis on CoCl2-treated cells compared with cells exposed to actual hypoxia. This comparison would have provided valuable insights into the similarities and differences between CoCl2-induced and hypoxia-induced gene expression changes in MDA-MB-231 cells. Future studies should aim to address this limitation by conducting RNA-sequencing analysis on both CoCl2-treated and hypoxic MDA-MB-231 cells. This study provides a more comprehensive understanding of the effects of CoCl2 on gene expression and its potential as a tool for studying hypoxia-related mechanisms in triple-negative breast cancer research.
4.Figure 3. The effect of the HIF-1alfa inhibitor on the wound-healing appears at too high concentration to be considered to be mediated by HIF.
Response:
Thank you for your question regarding the concentration of the HIF1α inhibitor used in Figure 3O. We apologize for the error in the figure, which incorrectly stated the concentration as 100 nM. The actual concentration used in the experiment was 50 nM. We understand your concern that the use of a higher concentration might not be mediated by HIF1α. However, we would like to clarify that the CCK8 assay in Figure 3K demonstrates that 50 nM of the HIF1α inhibitor does not induce cell death. Therefore, we believe that the observed inhibition of migration in Figure 3O is indeed mediated by HIF1α. We have corrected the error in the revised Figure 3O.
- 5. In the introduction and Discussion, it should be clarified that not all effects of hypoxia in tumors are mediated by HIF (see for example PMID: 38136613; PMID: 27028592; PMID: 25642170).
Response:
Thank you for your insightful comment and for highlighting the importance of considering non-HIF-mediated effects of hypoxia in our study. We agree that not all effects of hypoxia in tumors are mediated by HIF. We have clarified this point in the introduction and discussion sections of revised manuscript from page 3, line 97 to line 98, and from page 13, line 387 to line 388.
page 3, line 97 to line 98
Hypoxia also regulates migration/invasion and cell death by modulating ion channels, such as Ca2+, Cl- and K+ channels[36-38].
page 15, line 470 to line 472.
In addition to HIF1α, ion channels (K+ and Cl-) can lead to abnormal migration/invasion and resistance to cell death under hypoxia[36-38].
Comments on the Quality of English Language
The English needs to be significantly improved.
Response:
Thank you for your feedback regarding the English in the manuscript. We acknowledge the need for improvement and are committed to ensuring that the final version of the manuscript is written in clear, concise, and grammatically correct English.
Reviewer 4 Report
Comments and Suggestions for Authors
My decision: Major revision
Under hypoxic conditions, HIF-1α translocates into the nucleus to activate hypoxia-responsive genes. These genes are involved in processes such as apoptosis, angiogenesis, glycolysis, extracellular matrix remodeling, and cell cycle control. HIF-2 also contributes by modulating various widely expressed hypoxia-inducible genes. These changes promote the survival of malignant cells in the hypoxic microenvironment, resulting in a poorer prognosis. However, the precise mechanisms connecting hypoxia to tumor progression remain to be elucidated.
Major comments
Authors can enhance the content. However, What about the HIF-2 in these TNBCs upon hypoxia ? any note on it. ?
Refer: Knowles HJ, Harris AL. Hypoxia and oxidative stress in breast cancer. Hypoxia and tumourigenesis. Breast Cancer Res. 2001;3:318–22.
Hong SS, Lee H, Kim KW. HIF-1alpha: a valid therapeutic target for tumor therapy. Cancer Res Treat. 2004;36:343–53.
Hu CJ, Wang LY, Chodosh LA, Keith B, Simon MC. Differential roles of hypoxia-inducible factor 1alpha (HIF-1alpha) and HIF-2alpha in hypoxic gene regulation. Mol Cell Biol. 2003;23:9361–74.
Minor comments
English editing is required.
represent cells with arrow marks in Figure 3B, G
Check for plagiarism limit
Comments on the Quality of English LanguageSatisfactory
Author Response
Overall Response: We thank all reviewers for their valuable time and insightful comments on our manuscript. We appreciate the constructive feedback you have provided, which has helped us to improve the quality of our work. In this point-by-point response letter, reviewers’ comments were marked in dark blue italics, followed by our point-by-point responses. All revised or supplementary new text is underlined in red.
Comments and Suggestions for Authors
My decision: Major revision
Under hypoxic conditions, HIF-1α translocates into the nucleus to activate hypoxia-responsive genes. These genes are involved in processes such as apoptosis, angiogenesis, glycolysis, extracellular matrix remodeling, and cell cycle control. HIF-2 also contributes by modulating various widely expressed hypoxia-inducible genes. These changes promote the survival of malignant cells in the hypoxic microenvironment, resulting in a poorer prognosis. However, the precise mechanisms connecting hypoxia to tumor progression remain to be elucidated.
Major comments
Authors can enhance the content. However, What about the HIF-2 in these TNBCs upon hypoxia? any note on it.?
Refer: Knowles HJ, Harris AL. Hypoxia and oxidative stress in breast cancer. Hypoxia and tumourigenesis. Breast Cancer Res. 2001;3:318–22.
Hong SS, Lee H, Kim KW. HIF-1alpha: a valid therapeutic target for tumor therapy. Cancer Res Treat. 2004; 36:343–53.
Hu CJ, Wang LY, Chodosh LA, Keith B, Simon MC. Differential roles of hypoxia-inducible factor 1alpha (HIF-1 alpha) and HIF-2 alpha in hypoxic gene regulation. Mol Cell Biol. 2003; 23:9361–74.
Response:
Thank you for your insightful comment and for highlighting the importance of considering HIF-2 in our study of TNBCs under hypoxic conditions. We agree that HIF-2 is a crucial player in the hypoxic response and could have significant implications for TNBCs[39-41]. Kyoto Encyclopedia of Genes and Genomes (KEGG) pathway analysis showed that the upregulated DEGs were involved in pathways are mainly HIF-1 signaling pathway (Figure 2D). We analyzed the TFs of the 411 upregulated genes under hypoxia and found that these genes were regulated mainly by HIF1a and SP1 (Table 1). We also found that HIF-1α mRNA is upregulated 4-fold under hypoxic conditions, whereas HIF-2α mRNA does not appear to be upregulated in our RNA-sequencing data (Revised Figure 1B). It is possible that the cell line (MDA-MB-231) used in our study has a different regulatory mechanism compared to other TNBC cell lines. We will investigate the potential role of HIF-2α in mediating the effects of hypoxia on ROS production, cell migration, and other relevant processes. This may involve using HIF-2α inhibitors or siRNA to knockdown HIF-2α expression and then measuring the effects on these processes. We will also compare HIF-2α expression in our TNBC cell line to other TNBC cell lines to determine if there are any cell-type specific differences in HIF-2α expression or regulation in the future. We have now compiled the data into revised Figure S1B and described them in revised text from page 15, line 442 to line 453.
page 15, line 442 to line 453.
HIF2α is a crucial player in the hypoxic response and could have significant implications for TNBCs[39-41]. Thus, we compared the FPKM values of HIF1α and HIF2α (Figure S1A) and found that HIF-1α mRNA was upregulated 4-fold under hypoxic conditions, whereas HIF2α mRNA did not appear to be upregulated in our RNA-sequencing data. Kyoto Encyclopedia of Genes and Genomes (KEGG) pathway analysis revealed that the upregulated DEGs were involved in pathways related mainly to the HIF-1 signaling pathway (Figure 2D). The TFs of the 411 genes whose expression levels were upregulated under hypoxia were regulated mainly by HIF1a and SP1 (Table 1). The regulatory mechanism of the MDA-MB-231 cell line used in our study may differ from those of other TNBC cell lines. We will investigate the potential role of HIF2α in mediating the effects of hypoxia on MDA-MB-231 cells in the future.
Minor comments
1.English editing is required.
Response:
Thank you for your feedback regarding the English in the manuscript. We acknowledge the need for improvement and are committed to ensuring that the final version of the manuscript is written in clear, concise, and grammatically correct English.
2.represent cells with arrow marks in Figure 3B, G
Response:
Thank you for your careful review and insightful comments. We appreciate you bringing the suggestion to highlight ROS-producing cells to our attention. We have added white arrows to Figure 3B and Figure 3G to indicate the typical cells that are producing ROS (Revised Figure 3B, 3G), and described them in revised text from page 11, line 315 to line 316.
page 11, line 315 to line 316.
The white arrows in B and G represent cells that produce ROS.
3.Check for plagiarism limit
Response:
Thank you for your feedback regarding the plagiarism in our manuscript. We sincerely apologize for any instances of unintentional plagiarism and are committed to ensuring that the final version of the manuscript is entirely original and free of plagiarism.
Comments on the Quality of English Language
Satisfactory
Response:
Thank you for your careful review and positive comments.
Round 2
Reviewer 1 Report
Comments and Suggestions for Authors
The extensive revsion of the manuscript, including new sets of experiments, methods description, GEO repository, and title change render it duitable for publication.
Comments on the Quality of English LanguageAcceptable
Reviewer 3 Report
Comments and Suggestions for Authors
The Authors have satisfactorily addressed the major points of my criticism
Comments on the Quality of English LanguageEnglish improved
Reviewer 4 Report
Comments and Suggestions for Authors
Authors addressed all my comments and would suggest the authors to perform at least 3 times proofreading during proof stage and proceed for publication. At least 3 times.